# Encoder-Agnostic Adaptation for Conditional Language Generation

## Abstract

Large pretrained language models have changed the way researchers approach discriminative natural language understanding tasks, leading to the dominance of approaches that adapt a pretrained model for arbitrary downstream tasks. However, it is an open question how to use similar techniques for language generation. Early results in the encoder-agnostic setting have been mostly negative. In this work, we explore methods for adapting a pretrained language model to arbitrary conditional input. We observe that pretrained transformer models are sensitive to large parameter changes during tuning. Therefore, we propose an adaptation that directly injects arbitrary conditioning into self attention, an approach we call pseudo self attention. Through experiments on four diverse conditional text generation tasks, we show that this encoder-agnostic technique outperforms strong baselines, produces coherent generations, and is data-efficient.

## 1 Introduction

Large-scale language models have been shown to dramatically improve the performance of natural language understanding (NLU) systems on a broad range of tasks (Peters et al., 2018; Devlin et al., 2018; Radford & Salimans, 2018; McCann et al., 2017). The dominant paradigm is to pretrain a self attention-based language model on a large corpus of unlabeled text and then finetune the language model with an additional task-specific classification head on supervised data. Optimizing the effectiveness of this approach has been the focus of much study (Houlsby et al., 2019; Wang et al., 2019; Chronopoulou et al., 2019).

Given the success of pretraining for NLU tasks, how can large language models best be adapted for conditional language generation? Ideally, one should only need to train a large language model once and then apply it as part of the decoder to a range of tasks with different source modalities (e.g., text, images, bits). In the encoder/decoder framework, a task-specific encoder can encode source information into a continuous vector. The central question is thus how to adapt a pretrained decoder to effectively utilize arbitrary source information from an encoder (*encoder-agnostic* adaptation).

Considering the high quality of samples from large languae models (Radford et al., 2019), it is natural to expect encoder-agnostic adaptation to improve the coherence and grammaticality of conditional text generation over training a decoder from scratch, even when the source modality is not text (e.g. image captioning or class-conditional generation). Unfortunately, past results indicate otherwise. Edunov et al. (2019) show, for example, that a straightforward extension of contextual representations (Peters et al., 2018) to the conditional generation setting actually *hurts* performance compared to a model without any pretraining. Other pretraining approaches for language generation (Song et al., 2019; Dong et al., 2019; Lample & Conneau, 2019) have demonstrated strong performance on text-to-text tasks, but these methods are constrained to tasks where the source is natural language and do not address the encoder-agnostic setting.

In this work, we consider several different approaches to the problem of encoder-agnostic adaptation. We first observe that standard adaptation approaches perform poorly on this task. We hypothesize that because these techniques require relearning significant parts of the network structure to inject contextual conditioning, they move the parameters too far from the pretrained values. In contrast, Radford et al. (2019) observe that even trivial conditioning with the original model produces reasonable zero-shot generations without finetuning.

These results motivate an approach that learns the correct conditioning to control the model's output, which we call *pseudo self attention*. The idea is to learn a task-specific encoder that injects pseudo history into a pretrained self attention model. Because self attention works with sets of any size, the model can immediately utilize or ignore this history. Finetuning adapts the model to this new input while training a task-specific encoder.

Experiments utilize the GPT-2 (Radford et al., 2019) transformer as a pretrained model. We consider four diverse generation tasks spanning a range of source modalities: class-conditional generation, document summarization, story generation, and image paragraph captioning. Across all tasks, we find that pseudo self attention outperforms the other pretraining methods and is the most consistent. As a practical tool, pseudo self attention improves performance compared to a baseline without pretraining by large margins without sacrificing adherence to the source, even for tasks with large amounts of supervised data. We further demonstrate that the approach is data-efficient and produces qualitatively more coherent outputs. Code is available at `https://github.com/anon37234/encoder-agnostic-adaptation`.

## 2 RELATED WORK

**Pretrained Decoder Transfer learning for NLG**   Natural language generation (NLG) tasks have a long history of incorporating unconditional language models with conditional input(Bahl et al., 1983; Koehn et al., 2003). These approaches traditionally use the noisy channel model (i.e., Bayes' rule), and $n$-gram models as the language model. Recent adaptations of these ideas include the Neural Noisy Channel (Yu et al., 2017) as well as "fusion" methods (Koehn et al., 2003; Gulcehre et al., 2015; Sriram et al., 2018; Stahlberg et al., 2018) in which the output logits of a language model and a conditional model are combined to calculate the output probabilities. We consider this class of transfer learning as a baseline in a preliminary experiment (see Section 4.1), but focus on alternative "deep" approaches that incorporate the language model weights as an integral part of the model instead of an add-on at the end. Along these lines, Ramachandran et al. (2017) propose a finetuning-based method for machine translation with LSTMs, in which some of the layers of the LSTM are initialized with pretrained language model weights. As their method is specific to LSTMs, however, it is incompatible with modern transformer architectures.

**Pretraining-Based Transfer Learning for NLG**   Zhang et al. (2019) use BERT in the encoder and decoder of a summarization model via a unique cloze generative process. They demonstrate strong summarization performance, but the value of pretraining relative to other model components is not clear, and the cloze process significantly reduces the practicality of the model. More related, Edunov et al. (2019) experiment with a representation-based approach for applying ELMo (Peters et al., 2018) to the source and target sides of a standard seq2seq model separately. Their approach consistently improves performance when applied to the source, but hurts performance when applied to the decoder. We consider such a representation approach as a baseline in this work.

Most recently, several studies experiment with BERT-like masking approaches that are compatible with natural language generation (Song et al., 2019; Dong et al., 2019; Lample & Conneau, 2019). While these works demonstrate impressive performance, they are constrained to text-to-text tasks because they do not have a way to handle arbitrary conditional information. Whereas these works study pretraining methods that optimize transfer for text-to-text tasks, our study considers the separate problem of adapting a fixed pretrained model to arbitrary source conditioning.

Concurrent with this work, Golovanov et al. (2019) propose a similar approach to pseudo self attention and report initial experiments with dialogue generation. This study complements ours with positive results on dialogue generation, though we aim for experimental data over a wide range of language generation tasks and input modalities and comparison to strong encoder-agnostic baselines.

## 3 METHODS

We assume that we have a pretrained language model, $p(\boldsymbol{y}) = p(y_1, \ldots, y_T; \theta)$, that the model is an autoregressive neural network, and that it is based on self attention to implement conditioning, i.e.,

$$\text{SA}(Y) = \text{softmax}((YW_q)(YW_k)^\top)(YW_v),$$

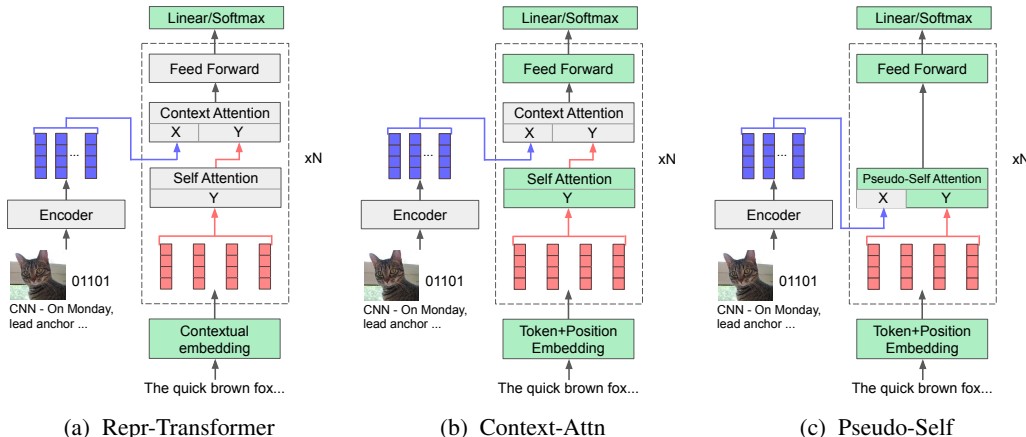

(a) Repr-Transformer      (b) Context-Attn      (c) Pseudo-Self

Figure 1: Encoder-agnostic variants considered. All methods utilize a task-specific source encoder, but vary in which parts of the decoder are pretrained and which are randomly initialized. Repr-Transformer trains a new full transformer decoder, Context-Attn trains a new context attention layer, Pseudo-Self attention only modifies part of the self attention layer. Residual connections and layer normalization have been omitted for clarity. Green indicates that parameters are initialized with pretrained weights, gray indicates random initialization. Red vectors indicate the target activations at each layer. Blue vectors indicate the source features at the output of the encoder. xN indicates that the section within the dotted lines is stacked N times.

where input $Y \in T \times D$ for hidden dimension $D$, $W_k, W_v, W_q \in D \times D'$ are parameters, representing the key, value, and query projections respectively, and the output is $T \times D'$. [1] We are interested in using this model to estimate the conditional probability $p(\boldsymbol{y} \mid \boldsymbol{x})$ for an arbitrary input $\boldsymbol{x}$ for which we have a small amount of supervised $(\boldsymbol{x}, \boldsymbol{y})$ pairs. The goal is to learn a model on this new data that best makes use of the pretrained model $p(\boldsymbol{y})$ with a method that is agnostic to the form of $\boldsymbol{x}$.

All models are based on the encoder/decoder architecture, and for each we follow the same high-level procedure: First, some of the weights of the decoder are initialized with weight values from a pretrained language model. Next, a task-specific encoder and all non-pretrained decoder weights are randomly initialized. Finally, the entire model is trained/finetuned end-to-end using the supervised data for the given task. In all cases, the input and output embeddings are tied. Each approach uses all of the pretrained weights, differing *only* in where and how they use pretrained weights in the decoder. Further experimental details are included in appendix Section A.

**Baseline 1: Repr-Transformer** The first approach considered (Fig 1a) utilizes the pretrained LM to produce a general-purpose representation of the target text before introducing the source information. For this method, a standard transformer decoder is used with the target word embeddings replaced by the output representation of the pretrained language model. In preliminary experiments, we considered both fixing and updating these representations and found that a fixed weighted-averaging ("ELMo-Style") method performed better, consistent with Edunov et al. (2019). One possible downside to this approach is that the conditioning information from the encoder is injected after all of the pretrained weights.

**Baseline 2: Context-Attn** The second approach (Fig 1b) considers initializing a standard transformer decoder with the shared weights of a pretrained LM. The newly added context attention weights at each layer are randomly initialized. Compared to Repr-Transformer, the conditioning information is injected alongside the pretrained weights. However, the randomly initialized context attention block may interfere with the carefully co-tuned pretrained weights of the rest of the model. This interference may introduce optimization challenges and lead to reduced performance.

---

[1]In practice many of these units ("heads") are stacked together via concatenation across dimension followed by a final linear projection $W_f \in D \times D$.

**Proposed Model: Pseudo-Self**   A more radical approach to incorporating conditional information is the "zero-shot" model proposed by Radford et al. (2019). Instead of learning a representation for $x$ and passing it into a context attention block they note that an auto-regressive model, $p(y_t \mid y_{<t})$, is already a conditional model. If $x$ is the same modality as $y$ (e.g., both language), one can condition on $x$ by prepending the source to target: $p(y_t \mid x, y_{<t}) = p(y_t \mid x \odot y_{<t})$.[2]  While this does not produce competitive models and is limited in its applicability, it is surprising that it works at all.

Taking inspiration from this approach, we propose learning this contextualization in an encoder-agnostic way. Our approach, pseudo self attention, simply injects learned encoder conditioning directly into the pretrained self attention of the model. Assume that we have a matrix $X \in S \times D$ representing a size $S$ encoding of $x$, define pseudo self attention as

$$\text{PSA}(X, Y) = \text{softmax}((YW_q) \begin{bmatrix} XU_k \\ YW_k \end{bmatrix}^\top) \begin{bmatrix} XU_v \\ YW_v \end{bmatrix},$$

where $U_k, U_v \in D \times D'$ are new parameters which project encoder outputs into the decoder self attention space. Because attention is inherently variable length, these additional inputs can be injected without changing the module and only act additively on the attention output. The full model is shown in Figure 1c.

Compared to Context-Attn, the proposed approach only introduces new parameters in the self attention block, which we expect leads to only minimal interference. As the pretrained LM weights encode for generation capability, deviating less from this initialization may lead to better generation performance. We explore this quantitatively in Section 5.

## 4   EXPERIMENTS AND RESULTS

Experiments consider four diverse tasks spanning input modalities, dataset sizes, and information about the target contained in the source. Tasks are chosen to emphasize long-form targets to probe the generation capabilities of the different models in a conditional setting. Perplexity is used to measure overall performance and diversity of output, combined with standard task-specific metrics.

For all tasks, we use GPT-2 small (117M parameters) (Radford et al., 2019) as the pretrained language model. GPT-2 small has 12 layers, 12 heads per layer, and a model dimension of 768 units; the Context-Attn and Pseudo-Self models use the same architecture. For the Repr-Transformer model to avoid overfitting, we use 6/8/512 layers/heads/dim for the decoder (in addition to the GPT-2 contextual representation network). All experiments use the same 50k type BPE GPT-2 vocabulary.

### 4.1   PRELIMINARY: CLASS-CONDITIONAL GENERATION

We first consider a control experiment with a minimal encoder model. We consider producing class-conditional samples, e.g., $p(y \mid x = 0)$ and $p(y \mid x = 1)$, from the IMDb sentiment classification dataset (Maas et al.), similar to previous works for sentiment transfer (Shen et al., 2017; Zhao et al., 2018). We set $x$ to be a sentiment bit (positive/negative), and the movie review as the target $y$. We maintain the original IMDb 25k/25k train/test split, with 2.5k reviews of the original training split held out for validation, and truncate reviews to 400 BPE tokens during training. Model quality is evaluated by perplexity, and adherence to the source bit $x$ is evaluated by the sentiment classification accuracy of an external classifier on generated reviews as in Shen et al. (2017). Reviews are generated via random sampling with a temperature of 0.7. For our external classifier we use fastText (Joulin et al., 2016), which has an accuracy of 90.1% on the IMDb test set.

Table 1 shows results for the conditional models, GPT-2 without finetuning, and Simple Fusion (Stahlberg et al., 2018). The GPT-2 model itself already shows a greatly reduced PPL compared to a baseline transformer. All pretraining methods further improve perplexity. The pseudo self attention approach significantly outperforms the approaches in terms of class adherence. Despite being initialized as a language model, the approach only sees a decrease of 0.4% classification accuracy compared to the randomly initialized model. In contrast, Repr-Transformer and Context-Attn see a decrease of 20.0% and 3.9%, respectively. We additionally report the results of Simple Fusion in

---

[2]This method is most successful when hand-selected task-dependent buffer words are inserted between $x$ and $y_{<t}$ as well such as "tl;dr" for summarization.

| Model | PPL ↓ | Cls Acc ↑ |
|---|---|---|
| Test set | - | 90.1 |
| GPT-2 | 41.21 | - |
| Simple Fusion | 38.31 | 65.1 |
| Transformer | 105.43 | **92.7** |
| Repr-Trans | 39.69 | 72.7 |
| Context-Attn | 40.74 | 88.8 |
| Pseudo-Self | **34.80** | 92.3 |

Table 1: Class-Conditional Generation on IMDb movie reviews. Classification accuracy is measured by a sentiment classifier trained on the IMDb training set. Bold indicates statistically significant best results at $p \leq 0.05$.

| Model | R1 ↑ / R2 ↑ / RL ↑ | PPL ↓ |
|---|---|---|
| PointerGen+BU | 41.22 / 18.68 / 38.34 | - |
| ELMo+SHDEMB[†] | 41.56 / 18.94 / 38.47 | - |
| BERT+Two-Stage[†] | 41.38 / 19.34 / 38.37 | - |
| UniLM+ExtLoss[†] | **43.47 / 20.30 / 40.63** | |
| Transformer+Copy | 39.94 / 17.73 / 37.09 | 8.21 |
| Repr-Trans | 37.09 / 13.77 / 33.99 | 13.58 |
| Context-Attn | 40.59 / 18.17 / 37.24 | 6.68 |
| Pseudo-Self | 40.72 / **18.38** / 37.46 | **6.43** |
| Pseudo-Self+BU | **41.62** / 18.66 / 38.46 | **6.43** |

Table 2: Abstractive summarization on CNN/DM. † indicates pretraining of the encoder side. PointerGen+BU from (Gehrmann et al., 2018), ELMo+SHDEMB from (Edunov et al., 2019), BERT+Two-Stage from (Zhang et al., 2019), UniLM+ExtLoss from (Dong et al., 2019). Bold indicates statistically significant best results among general models and encoder-agnostic models at $p \leq 0.05$.

Table 1. Compared to Pseudo-Self, it gives a worse PPL and inferior classification accuracy. Given the weak results, we focus on comparisons between the deep models for the rest of the paper.

## 4.2 DOCUMENT SUMMARIZATION

Abstractive document summarization requires the model to produce a long-form summary given a full news article. For these experiments, we use the non-anonymized CNN-Daily Mail dataset (Hermann et al., 2015). The dataset is comprised of 280k training examples of document-scale source news articles and corresponding 2-4 sentence target summaries. Summarization is a mature testbed with state-of-the-art models that use task-specific architecture modifications, so transfer learning methods need to be able to mesh well with these changes. We use the transformer version of the copy mechanism from Gehrmann et al. (2018) and employ bottom-up (BU) summarization attention pruning (Gehrmann et al., 2018). Generation is conducted via beam-search with a beam size of 5 with tri-gram blocking, consistent with the literature models (Edunov et al., 2019).

Table 2 shows the performance of the models tested with recent state-of-the-art models for comparison. Compared to the baseline model without pretraining, Pseudo-Self improves ROUGE-1 by 0.78, ROUGE-2 by 0.65, ROUGE-L by 0.37, and reduced PPL by 20%. The Context-Attn approach nearly matches these results for this task, but the Repr-Transformer approach performs more poorly.

We additionally experiment with the bottom-up summarization attention pruning approach applied at inference time as in Gehrmann et al. (2018). With this modification, Pseudo-Self outperforms all literature models in ROUGE-1 except the text-to-text UniLM+ExtractLoss, which uses joint pretraining of the source and target and is trained with an additional extractive loss. The performance of all of our models can potentially be further improved through a pretrained encoder.

## 4.3 CONDITIONAL STORY GENERATION

Conditional story generation with the WritingPrompts dataset (Fan et al., 2018) requires the model to produce an on-topic story given a short textual prompt. While summarization relies heavily on the encoder, this task gives more flexibility to the decoder. The dataset is well supervised, containing 300k single sentence writing prompts (the source) and stories (the target). Following the preprocessing of Fan et al. (2018), we truncate the stories to 1000 tokens. Due to the story lengths, the total number of training tokens is on the order of 100 million, resulting in a large in-domain data setting.

To compare models, we compute two metrics: perplexity (PPL) and prompt ranking. Perplexity assess approximate quality and diversity, whereas prompt ranking measures the relevance of the story to the prompt. To calculate the prompt ranking, we use the procedure from Fan et al. (2018): For each story in the test set, the likelihood is evaluated under the model for the "true" corresponding

| Model | PPL ↓ | Rank Acc. ↑ |
|---|---|---|
| Transformer | 30.58 | 80.6 |
| Repr-Trans | **21.16** | 76.7 |
| Context-Attn | >5000 | 9.3 |
| Pseudo-Self | 21.21 | **81.8** |

Table 3: Story generation on the Writing-Prompts dataset. *Rank acc.* refers to the top-1 prompt ranking accuracy metric described in Section 4.3. (Experiments use the GPT2 BPE scheme, so PPL numbers are not directly comparable to those reported in (Fan et al., 2018)). Bold indicates statistically significant best results at $p \leq 0.05$.

| Model | CIDEr ↑ | B4 ↑ |
|---|---|---|
| Krause et al. (2017) | 13.5 | 8.7 |
| Chatterjee et al. (2018) | 20.9 | **9.4** |
| Melas-Kyriazi et al. (2018) | 22.7 | 8.7 |
| Transformer | 19.9 | 8.0 |
| Repr-Trans | 19.3 | 7.2 |
| Context-Attn | **22.6** | 7.6 |
| Pseudo-Self | **24.0** | 8.3 |

Table 4: Image paragraph captioning on Visual Genome, as measured by *CIDEr* and *BLEU-4* (B4) scores. Bold indicates statistically significant best results at $p \leq 0.05$.

prompt and 9 other randomly selected "fake" prompts from the test set. Then, the rank accuracy is the percentage of stories for which the model gave the highest likelihood to the correct prompt.

Table 3 shows the results. Despite the large dataset size, the Repr-Transfomer and Pseudo-Self approaches still substantially reduce the PPL, suggesting that these models effectively make use of the GPT-2 LM. Pseudo-Self sees only a 0.3% decrease in prompt ranking accuracy, while the Repr-Transformer approach sees a larger decrease. The Context-Attn model runs into optimization challenges and fails to learn in this setting. We hypothesize that this failure is a result of introducing a randomly initialized attention block, which makes Context-Attn susceptible to optimization challenges, but further work is needed to understand this more completely.

### 4.4 IMAGE PARAGRAPH CAPTIONING

Our final set of experiments consider image paragraph captioning using the *Visual Genome* dataset from Krause et al. (2017). Image captioning represents a strong real-world use case for encoder-agnostic pretraining. Visual Genome, in particular, represents a realistic setting with paragraph-sized captions (5-8 short sentences), which requires greater fluency than single sentence captions. Due to the difficulty of producing labeled paragraph captions Visual Genome, contains fewer than 20,000 image-paragraph pairs. As a result, models trained from scratch on Visual Genome have been observed to have difficulty learning the structure of language.

We use the same convolutional encoder as Krause et al. (2017), without the final pooling layer: for each image the output of the encoder is a tensor of size $(36, 2048)$ extracted from a ResNet. Note that in this experiment the encoder and decoder are trained separately rather than end-to-end. Models are evaluated using the common CIDEr and BLEU-4 metrics.

Table 4 shows the results on the captioning task, comparing the transfer learning methods with a non-pretraining baseline and models from the literature which use the same loss function[3]. Of the three pretraining approaches, Pseudo-Self and Context-Attn give the statistically significant best performance, and Pseudo-Self is the only model to improve both CIDEr and BLEU compared to the Transformer baseline. Pseudo-Self additionally improves performance over the literature models in terms of CIDEr but gives a slightly worse BLEU-4.

## 5 ANALYSIS AND DISCUSSION

**Experimental trends** Overall, Repr-Trans gives poor performance in three out of the four tasks, underperforming a transformer without pretraining on summarization and image captioning. Context-Attn gives stronger results than Repr-Trans, but shows significantly worse performance than Pseudo-Self on class-conditional generation and slightly worse performance on summarization

---

[3]Recent work shows it is possible to improve paragraph captioning models by incorporating sequence-level (Melas-Kyriazi et al., 2018) and adversarial (Chatterjee & Schwing, 2018) losses, but these loss function improvements are orthogonal to improvements in the underlying model architecture.

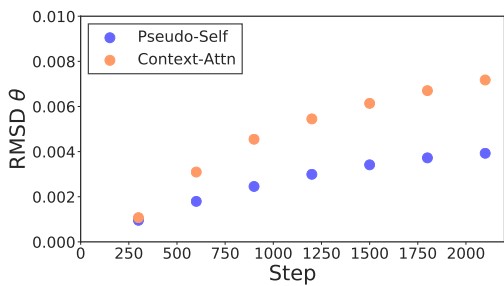

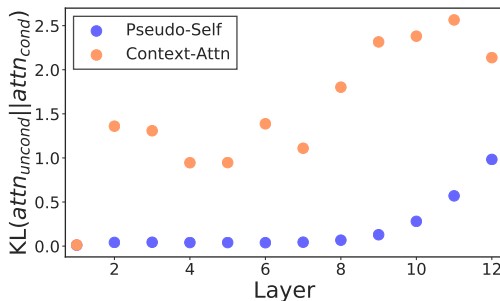

(a) Comparison of parameter changes in feed forward layers with different conditioning. Root median squared deviation between feed forward parameters, for the Pseudo-Self and Context-Attn models. The Context-Attn approach requires a larger deviation from the initialization to fit the data.

(b) KL divergence between the self attention distributions of a finetuned unconditional model and the self attention distribution of Pseudo-Self and Conext-Attn conditional models. Averaged over heads and location in target, computed at the end of training on the test target-side data.

Figure 2: Effect of introducing randomly initialized parameters.

and image captioning. Most critically, Context-Attn demonstrates a susceptibility to optimization challenges. On all tasks Pseudo-Self gives the best performance or is tied for best.

**Interference by added parameters** In Section 3, we hypothesized that pseudo self attention enables better use of the pretrained LM because the introduction of parameters in Pseudo-Self interferes less with the original parameters than Context-Attn. To explore this quantitatively, we plot the root median squared deviation of parameters from their original values in the feed-forward layer for the class-conditional generation task (Figure 2a). While both models start with the same parameters, the Context-Attn parameters change significantly more than Pseudo-Self during training.

The parameter values are only part of the story: parameters may change while the overall generative behavior stays the same. We probe this for image captioning by first finetuning the pretrained LM on target-side data, giving an unconditional model with an identical structure to Pseudo-Self and Context-Attn except without any additional randomly initialized parameters. In Figure 2b, we plot the KL divergence between the self attention distributions of the unconditional model and those of the conditional models at each layer. Both models have similar attention distributions to the unconditional model at the first layer (in Context-Attn this precedes the introduction of new parameters). Beyond the first layer, the KL for Context-Attn becomes over an order of magnitude larger than that of Pseudo-Self, suggesting that the introduction of the context attention block significantly perturbs the overall behavior of the model. The additional perturbation is not associated with improved incorporation of the source information, as Table 4 shows that Pseudo-Self gives better performance.

**Effect of pretrained LM size** There is a continuing trend to larger pretrained LMs. During the preparation of this manuscript, a larger version of GPT-2 was made available with 345M parameters, increasing the model dimension to 1028, the number of attention heads to 16, and the number of layers to 24. We retrained our model using this larger LM for class-conditional generation, using the same training hyperparameters and re-tuning the generation temperature (Table 5). The larger model improves PPL by 4.5 points while attaining similarly high classification accuracy. This datapoint suggests that transfer learning effectiveness can continue to improve along with the quality of the pretrained model used.

| Model | PPL ↓ | Cls Acc ↑ |
|---|---|---|
| Pseudo-Self 117M | 34.80 | 92.3 |
| Pseudo-Self 345M | 30.26 | 92.4 |

Table 5: IMDb conditional movie review generation results, comparing the larger 345M parameter GPT-2 model to the 117M parameter GPT model.

**Low-data supervision** Many of our tasks showed improvements even with medium-to-large training sets. To study the effectiveness of the approach in low data regimes, we create small datasets by subsampling the IMDb dataset to sizes between 200 and 16k datapoints. We retrain our model using the same hyperparameters and use datasize-dependent early stopping to prevent overfitting.

| Model | Grammaticality | Non-redundancy | Consistency | Typicality | Combined |
|---|---|---|---|---|---|
| Test set | $71.3 \pm 4.3$ | $87.2 \pm 3.2$ | $85.1 \pm 3.4$ | $74.4 \pm 4.1$ | $3.18 \pm 0.10$ |
| Transformer | $55.4 \pm 4.7$ | $60.5 \pm 4.6$ | $53.7 \pm 4.7$ | $39.7 \pm 4.6$ | $2.09 \pm 0.13$ |
| Repr-Trans | $\mathbf{62.1} \pm 4.4$ | $\mathbf{71.0} \pm 4.1$ | $\mathbf{57.1} \pm 4.5$ | $\mathbf{43.7} \pm 4.5$ | $\mathbf{2.34} \pm 0.12$ |
| Pseudo-Self | $\mathbf{65.2} \pm 4.6$ | $\mathbf{69.3} \pm 4.5$ | $\mathbf{61.3} \pm 4.7$ | $\mathbf{48.4} \pm 4.8$ | $\mathbf{2.44} \pm 0.13$ |

Table 6: Human evaluation of story generation quality. Participants were asked specific binary questions concerning the four criteria, the numbers for the four left categories represent percentages of approval. On the right, the methods are rated on a 4-point scale based on the combination of the four criteria. Uncertainties represent a 95% confidence interval, bold indicates statistically significant maxima for each category of the models under consideration.

To reduce variance and measure uncertainty, we repeat the process 8 times for each dataset size, calculating the PPL and classification accuracy. Results are shown in Figure 3. Note that a non-pretrained model has a PPL of over 1000 when trained on 200 examples. The pretrained model starts with reasonable outputs (44.4 PPL after 200 examples) and increases task accuracy steadily with more data. See Section B in the appendix for representative samples.

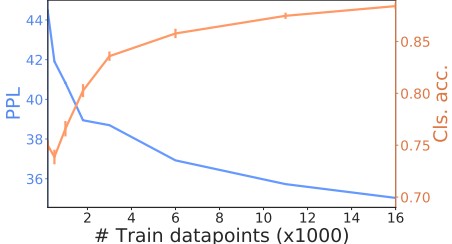

Figure 3: Data efficiency analysis with IMDb. PPL shown in blue (left), classification accuracy shown in orange (right). Error bars show an approximate 95% confidence interval.

**Human evaluation** To assess the quality of generations, we conducted a human evaluation based on the story generation task. Generation uses a temperature of 0.9 and a top-k value of 100. We ask participants on Amazon Mechanical Turk a series of four yes/no questions mapped to desirable linguistic properties outlined in Dang (2006): grammaticality, non-redundancy, consistency, and typicality. 125 stories are evaluated for each model, and each story is evaluated by 5 unique workers. Scores are calculated for each property as the total percentage of positive responses. A combined score rates the model overall on a scale from 0-4 based on the equally-weighted combination of the four properties.

The results are shown in Table 6. In all four categories, the Pseudo-Self and Repr-Transformer models show statistically significant performance gains compared to the baseline Transformer model. The Pseudo-Self model achieves a grammaticality score of only 6.1% less than the test set, indicating strong grammaticality, likely a more localized property, is well learned by the pretrained LM and effectively transferred to the conditional models. In contrast, all models score significantly worse than the test data in terms of consistency and typicality. This suggests that these higher-level properties, while best transferred in the Pseudo-Self case, still represent a challenge for neural models.

# 6 CONCLUSION

We study encoder-agnostic approaches for adapting a pretrained language model to general-purpose conditional language generation. Experiments spanning a range of diverse long-form conditional generation tasks demonstrate that pseudo self attention improves performance over strong encoder-agnostic pretraining baselines, and is the only consistently performant model. From a practical perspective, the approach gives robust, sizable improvements over a non-pretraining baseline while maintaining adherence to the source context. Furthermore, we demonstrate the data efficiency and qualitative properties of the approach.

Beyond empirical results, this study highlights the distinction between improving our ability to produce contextual representations of a source language and improving our capacity to generate text in a target language. While they appear to be similar problems, they exhibit substantially different phenomenology. For example, the representation-based approach, which works well for NLU, gives poor performance for NLG. Future work can study this distinction further.

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

## A    EXPERIMENTAL DETAILS

Approximate hyperparameter settings were taken from Gehrmann et al. (2018), followed by a coarse hyperparameter tuning for each dataset. Most parameters were held constant between the different models, though in some cases to enable as fair a comparison as possible individual parameters were separately optimized for each model. For the complete set of hyperparameters for each model, see the details at `https://github.com/anon37234/encoder-agnostic-adaptation`.

For all models Dropout and early stopping are used for regularization. In addition, in initial experiments we found two optimization details to help generalization across all models: discriminative finetuning (Howard & Ruder, 2018) and using a lower learning rate for the decoder than the encoder. In discriminative finetuning, the learning rate of each layer decreases exponentially from the top transformer layer to the bottom transformer layer. Using a smaller learning rate for the decoder ensures that the decoder does not initially significantly change to accommodate the uninformative information from the randomly initialized encoder.

## B    QUALITATIVE EXAMPLES

Representative samples for the movie review dataset are shown in Table 7. The No-Pretraining model is the transformer from Table 1 and the number in the left column indicates the number of supervised examples in the training dataset. Samples are generated via random sampling with a temperature of 0.75. Without pretraining, the model makes many coherence mistakes. The Pseudo-Self 22K makes no grammatical mistakes and follows a single train of thought, although it is somewhat generic.

The distinction between the models is further exaggerated when only 1.8k supervised examples are given. The baseline model trained on only 1.8k datapoints leads to very incoherent generated text. In contrast, the Pseudo-Attention model shows significantly improved grammar and sentence structure. Despite a handful of mistakes, the review follows a consistent description of a movie over multiple sentences. Given the poor performance of the baseline model, these properties must have been transferred from the original unconditional LM. These samples were selected to be representative of the broader set for the indicated models.

| Sentiment: | Positive |
|---|---|
| **No pretraining 1.8k:** | "Set's that I liked this movie. I have seen I remember the original movie is one of the music that it is great movie. I've seen this film and one of the whole movie is like this movie. It is so bad, I watched the top of this movie. i would see the movie was bad, I have seen it. This movie, it's a TV main movie is about the plot, relaxing. I liked this movie takes it is a few times, was awesome. I'm a DVD. The critics in the first time I know it were a lovely plot. You could watch I'm seen the acting, and I don't watch this. I've seen what you really i love the film. The film on the DVD. |
| **No pretraining 22k:** | I was enthralled by the performance of Hartnett at the time of this film. At the time this film was shot, I thought, I was absolutely fantastic in the film, and thought it was a real ride. It is about a family living in a small town San Francisco, where a couple lives in Mexico, where they are working at home with the family. In the end, they are in the family to decide the family will just have their fathers through endless periods, coffee, disillusionment, and finally, can't escape, but there's no story. There are some delightful performances (particularly the two leads) and the actors are well known, I really can't help but feel that the film could have been better as a comedy, with the exception of Ernesto L'Amelie. |
| **Pseudo-Self 1.8k:** | I fell in love with this film in 1985. It's a quintessential short film that explores the everyday lives of the human condition. The main character of the movie is a man named Donald (Husband George). He buys a home and captures a great deal of information about the businessmen who live and work in his neighborhood. This film is a complete departure from the norm or what a typical American would expect from a successful businessman with a short fuse. His stereotypical actors are less than likable, and his best friend is Kelly (Preet Bharara). The film is full of humor, and a great deal of heart and soul. |
| **Pseudo-Self 22k:** | When I saw the preview of this film, I thought it was going to be a horrible movie. I was wrong. The film has some of the funniest and most escapist scenes I've seen in a long time. The acting is superb. The story is decent, but the direction and editing may have been a bit harsh at times. The director and cast achieved a great balance of comedy and drama. I've seen some bad films, but this one is one of the ones I've seen that is really good. I loved the acting and the pace. The two leads were compelling. The only real problem with the film was that I was a bit bored with it. The ending is a bit long, but it's still a funny, good movie. It's efficient. I give it a 7/10. |

Table 7: Example generations from models trained on the movie review generation task. In all cases the indicated sentiment was positive. The number in the left column is the number of training examples (22k is the full dataset).

