# OpenReview forum: "Encoder-Agnostic Adaptation for Conditional Language Generation"
_ICLR.cc/2020/Conference — Reject_

### Official Review · AnonReviewer1 · 2019-10-21
**Official Blind Review #1**

**Rating:** 8

**Review:**

This paper proposes a simple yet effective method to adapt large-scale pre-trained language models, which have been shown to substantially improve performance on broadly classification-based NLU tasks, to NLG. The approach is explored in the encoder-agnostic {X}-to-text setup, where the source encoding {X} could represent arbitrary modalities, such as text or images.

More concretely, the paper leverages a pre-trained, large-scale language model (in this case a GPT-2), and examines how to best cast such unconditional language model into a decoder that generates text conditional on the source information {X}. As self-attention inherently works with sequences of any length, the proposed pseudo self-attention approach simply injects the encoder representation as additional conditioning context (using some additional projection matrices that are learned from scratch) into the pre-trained self-attention layers of the decoder. Extensive experiments and analysis on four diverse tasks demonstrate that pseudo self-attention generally outperforms two other ways of pre-training the decoder, improves NLG data efficiency, and produces texts that are judged more favourably by human evaluators.

Overall, this paper presents a simple, general, and effective method for adapting large-scale pre-trained language models to conditional text generation. Based on the pros and cons that I have listed below, I am giving a rating of "Accept". I hope that some of my concerns will be addressed in the authors' response.

Pros:
1. The paper is well-written and the methodology is explained very clearly. Figure 1 is particularly helpful in illustrating the differences between pseudo self-attention and the baselines.

2. The paper addresses a very important problem, and helps make sure that the advances that have been made in language modelling (which can leverage large amounts of unlabelled data), would transfer well to conditional text generation tasks, which hold immediate practical value yet often require expensive annotations.

3. The approach is simple and easy-to-implement, but has been shown to be effective across a broad range of problems, multiple modalities, and various evaluation metric.

4. The paper features extensive reference to relevant prior work, and clearly highlights the key similarities and differences with prior approaches.

Cons:
1. It is still unclear how using language model pre-training affects adequacy (as opposed to fluency). The paper shows that using pseudo self-attention results in a decoder that diverges less from its language model initialisation. One potential risk is that the decoder may prefer fluent, "safe" outputs (which is arguably what a language model would prefer since it is an unconditional model) that are nevertheless less faithful to the source information. Since none of the evaluation metric specifically assesses for adequacy on its own, it would be good to isolate the effect of pseudo self-attention on adequacy, and compare it with the baselines, in addition to a Transformer trained from scratch on each downstream task. How to measure adequacy is naturally still an open question, but there are a few things that can be done (e.g. recall of salient information, reverse perplexity to see how much of the source information can be "reconstructed" given the predicted target text, etc.).

2. It would be interesting to further examine the interaction between encoder pre-training and the decoder pre-training that is explored in this work. Another interesting experiment to run is whether end-to-end training (including fine-tuning the encoder) would help, since prior work has shown the benefits of end-to-end learning (at least when large amounts of data are available).

Questions:
1. Why is the context-attn model performance not included in Table 6? Is it because of the optimisation issue associated with that model in Table 3?

2. In page 7, it is mentioned that "Both models have similar attention distributions ... at the first layer, which precedes the introduction of new parameters in both models". Does the first layer here refer to the token + position embedding layer?

**Experience Assessment:**

I have read many papers in this area.

**Review Assessment: Checking Correctness Of Derivations And Theory:**

I assessed the sensibility of the derivations and theory.

**Review Assessment: Checking Correctness Of Experiments:**

I assessed the sensibility of the experiments.

**Review Assessment: Thoroughness In Paper Reading:**

I read the paper at least twice and used my best judgement in assessing the paper.

---

> ### Author Response · Authors · 2019-11-12
> **Response**
>
> We thank the reviewer for their positive comments and their questions.
>
> In response to the reviewer’s cons:
>
> 1. We agree that it can be challenging to measure the adequacy of language generation, as empirically PPL correlates more strongly with fluency. In the paper, we have isolated this specifically with the classification accuracy metric for class-conditional generation (Table 1) and the rank accuracy metric for story generation (Table 3). For the former, we use an external classifier trained on IMDb and measure the classification accuracy of generated reviews. If the model did not adhere to the source bit the external classifier would report a low classification accuracy. From Table 1 we see that, as you note, Repr-Transformer and Context-Attn show worse adherence to the source than the baseline trained from scratch. In contrast, Pseudo-Self is both fluent and adheres as well to the source as the baseline. We find similar behavior with story generation, as measured by the ability of the models to give the highest conditional likelihood p(y|x) for a fixed y to the true x.
>
> 2. Exploring the interaction between encoder pre-training and decoder pre-training is a very  interesting avenue for future research. We agree that when large amounts of data are available in a given setting (e.g. machine translation), pre-training both the encoder and decoder in an end-to-end manner is most effective (Meng et al, 2019; Song et al, 2019). Here, we are primarily interested in isolating improvements in fluency coming from pretraining the decoder, for general text generation tasks.
>
> In response to the reviewer’s questions:
>
> 1. Yes, you are exactly right -- the context-attn model performance not included in Table 6 because it did not produce coherent outputs, as shown in Table 3.
>
> 2. The first layer refers to the first self-attention block of the decoder (after the token + positional embedding layer). Thanks for bringing this up though, the first layer for pseudo-self does involve new parameters so this is misleading (the point is mainly about Context-Attn). We have updated the paper to reflect this.

---

> > ### Comment · AnonReviewer1 · 2019-11-13
> > **Reply**
> >
> > Thank you for clarifying this. I agree that the IMDb classification result is an evidence that the approach preserves adequacy, and the improvement comes not just from fluency.

---

### Official Review · AnonReviewer3 · 2019-10-23
**Official Blind Review #3**

**Rating:** 8

**Review:**

This paper proposes a new architecture to train decoder models on language generation using a pre-trained encoder (such as BERT or GPT-2). They introduce a novel block called `````"pseudo self-attention" that allow injecting the input for conditional generation in the self-attention layer (i.e. softmax of YW_q (XU_k | YW_k)^T (XU_v | YW_v) instead of softmax(YW_q(YW_k)^T)YW_v). They extensively evaluate their approach on a large set of tasks showing improvements across all of them (which includes class-conditional generation, summarization, story generation and paragraph generation). They also provide interesting ablation studies.

This paper proposes a simple architectural block to try and translate the success of large pre-trained encoders on discriminative tasks to the generative setting. The idea seems well-motivated and the paper is well-written and easy to follow. The experimental section is very thorough and show large improvements on a variety of task---I particularly appreciate that they experimented with conditional inputs of different nature (class value, image, different languages etc...) to show the effectiveness of their method.

Overall, while the idea is quite simple, the experiments speak for themselves and this could prove to be a useful `layer' to use on large pre-trained language models.

**Experience Assessment:**

I have published one or two papers in this area.

**Review Assessment: Checking Correctness Of Derivations And Theory:**

I assessed the sensibility of the derivations and theory.

**Review Assessment: Checking Correctness Of Experiments:**

I assessed the sensibility of the experiments.

**Review Assessment: Thoroughness In Paper Reading:**

I made a quick assessment of this paper.

---

> ### Author Response · Authors · 2019-11-12
> **Response**
>
> We thank the reviewer for their positive comments.
>
> We agree with the insight that our method could be adopted as a standard “layer” in large pre-trained language models.

---

### Official Review · AnonReviewer2 · 2019-11-02
**Official Blind Review #2**

**Rating:** 3

**Review:**

This paper compares a few encoder agnostic methods for using pretrained decoders in text generation tax. The author compared a few intuitive ways of doing this, and presents results showing that that pseudo-self attention does the best.

However, I think the results has some strange points that needs further investigation. Going from repr-transfomer to context-attention to pseudo-self, there is an increasing amount of parameters initialized by pretraining. However, both of the first two methods often perform worse than the baseline transformer without pretraining. So should more things be initialized with pre-training or less? It would be good to verify that this is not due to under-training.

Except paragraph captioning, the results on other tasks are not better than prior results, which do not use pretraining. The baseline transformer is also usually worse than prior results. The human evaluation shows that the proposed method do better on story generation, but this one is essentially text to text. What is missing is how this compares with even more pretraining, say GPT-2, without any fine tuning.

Transferring gains of pretraining to generation tasks is clearly a promising direction, and the bar for success in this area need to be outperforming the best previous methods that do not use pretraining.  There is no comparison with previous text 2 text methods that use pretraining.  If the proposed methods are truely encoder agnostic, then they should perform reasonably on text-to-text as well. I think some MT experiments would be good since the evaluations are more competitive and reliable. Perhaps using some language pairs that do not have sufficient training data.

**Experience Assessment:**

I have read many papers in this area.

**Review Assessment: Checking Correctness Of Derivations And Theory:**

I assessed the sensibility of the derivations and theory.

**Review Assessment: Checking Correctness Of Experiments:**

I assessed the sensibility of the experiments.

**Review Assessment: Thoroughness In Paper Reading:**

I read the paper at least twice and used my best judgement in assessing the paper.

---

> ### Author Response · Authors · 2019-11-12
> **Response**
>
> We thank the reviewer for their comments and suggestions.
>
> “Going from repr-transfomer to context-attention to pseudo-self, there is an increasing amount of parameters initialized by pretraining. However, both of the first two methods often perform worse than the baseline transformer without pretraining. So should more things be initialized with pre-training or less?”
>
> We agree that taking into account pretrained parameter counts is important. All of the methods we experiment with, repr-transformer, context-attention, and pseudo-self, use *exactly the same number of parameters* initialized by pretraining. Each uses all of the parameters of GPT-2, they just differ in how these parameters are incorporated into the conditional model. We have clarified this point in the updated version.
>
> It is interesting that the first two methods in some cases perform worse than the baseline without pretraining. As mentioned in the paper, this is consistent with previous results on similar representation-based approaches (Edunov et al. 2019). Because each approach uses the same pretrained parameters, this provides strong evidence for the central argument laid out in the paper that pretraining should be utilized but how the parameters are used in the conditional model is critical.
>
> “It would be good to verify that this is not due to under-training.”
>
> For all models we use early stopping and separately tune regularization parameters (dropout) on the validation set, and we have confirmed that the numbers we report in the paper are converged. We have clarified this in the appendix in the updated version.
>
> “Except paragraph captioning, the results on other tasks are not better than prior results, which do not use pretraining. The baseline transformer is also usually worse than prior results … Transferring gains of pretraining to generation tasks is clearly a promising direction, and the bar for success in this area need to be outperforming the best previous methods that do not use pretraining.”
>
> We are confused by this claim. Our results indicate that pseudo-self outperforms the state-of-the-art without pretraining.
>
> -For summarization, the best previous results without pretraining is PointerGen+BU, which pseudo-self outperforms.
> -For story generation, the best previous results use a standard seq2seq model. The results presented in this paper demonstrating that pseudo-self outperforms a baseline without pretraining represents an improvement of the state-of-the-art. (Due to differences in tokenization (we require using the GPT-2 tokenization) and handling unks it’s not possible to directly compare PPL results with Fan et al. so we compare instead with a comparable model using the GPT-2 tokenization. To verify that our baseline is indeed comparable, we reran what we call the “baseline transformer” in this paper using the exact tokenization from Fan et al. 2018. We get a PPL of 34.28; the best model in Fan et al. 2018 gets a PPL of 36.56.)
> -For paragraph captioning, pseudo-self outperforms previous models in terms of CIDEr.
> - For class-conditional generation a transformer is a very strong baseline without pretraining. All pretrianing approaches improve PPL from ~100 PPL to ~35 PPL, and pseudo-self improves PPL without sacrificing adherence to the source.
>
>
> “What is missing is how this compares with even more pretraining, say GPT-2, without any fine tuning.”
>
> Could you clarify what you mean by this? GPT-2 is the pretrained model we use in the paper, and in each model we use all of the parameters from GPT-2.
>
> “There is no comparison with previous text2text methods that use pretraining.  If the proposed methods are truely encoder agnostic, then they should perform reasonably on text-to-text as well.”
>
> We agree that encoder-agnostic methods should perform well on text2text tasks, ideally comparable to non-encoder-agnostic pretraining methods. While the focus of this paper is pretraining on the decoder side, in the summarization text2text task the literature baselines reported in Table 2 include BERT+Two-Stage (Zhang et al., 2019) and UniLM+ExtLoss (Dong et al., 2019) which use pretraining approaches specialized for text2text in both the encoder and decoder. We obtain similar results to BERT+Two-Stage (outperform in R1 and RL), with a much simpler architecture. We obtain worse results than UniLM, which includes a number of orthogonal additions including an extractive loss, joint pretraining of the source and target specific for text2text tasks, and a larger pretrained model. Overall, while the approach is not optimized for text2text tasks the results demonstrate reasonable performance where comparable, with the major benefit of being encoder-agnostic.
>
> “I think some MT experiments would be good since the evaluations are more competitive and reliable. Perhaps using some language pairs that do not have sufficient training data.”
>
> This is a good suggestion to augment our current text2text results, and we will pursue it in the future.

---

### Decision · Program_Chairs · 2019-12-19

**Decision:**

Reject

**Comment:**

This paper proposes a method to use a pretrained language model for language generation with arbitrary conditional input (images, text). The main idea, which is called pseudo self-attention, is to incorporate the conditioning input as a pseudo history to a pretrained transformer. Experiments on class-conditional generation, summarization, story generation, and image captioning show the benefit of the proposed approach.

While I think that the proposed approach makes sense, especially for generation from multiple modalities, it would be useful to see the following comparison in the case of conditional generation from one modality (i.e., text-text such as in summarization and story generation). How does the proposed approach compare to a method that simply concatenates these input and output? In Figure 1(c), this would be having the encoder part be pretrained as well, as opposed to randomly initialized, which is possible if the input is also text. I believe this is what R2 is suggesting as well when they mentioned a GPT-2 style model, and I agree this is an important baseline.

This is a borderline paper. However, due to space constraint and the above issues, I recommend to reject the paper.